# Symptomatic Popliteal Artery Aneurysms in Recently SARS-CoV-2-Infected Patients: The Microangiopathic Thrombosis That Undermines Treatment

**DOI:** 10.3390/diagnostics13040647

**Published:** 2023-02-09

**Authors:** Laura Capoccia, Wassim Mansour, Luca di Marzo, Sabrina Grimaldi, Alessia Di Girolamo

**Affiliations:** 1Department of Surgery, Vascular Surgery Division, SS Filippo e Nicola Hospital, 67051 Avezzano, Italy; 2Department of General Surgery and Specialist Surgery, Sapienza University of Rome, Policlinico Umberto I of Rome, Viale del Policlinico 155, 00162 Rome, Italy

**Keywords:** popliteal artery aneurysm, endovascular treatment, COVID-19, thrombosis, femoro-popliteal bypass

## Abstract

Background: Arterial and venous thrombosis are complications in SARS-CoV-2-infected patients. The microangiopathic thrombosis in affected patients can compromise results in urgent limb revascularizations. Aim of our study is to report on the incidence of symptoms development in patients affected by popliteal artery aneurysm (PAA) and to analyze the effect of COVID-19 infection on outcomes. Methods: Data on patients surgically treated for PAA from the massive widespread of COVID-19 vaccine (March 2021) to March 2022 were prospectively collected. Factors considered for analysis were: presence of symptoms, aneurysm diameter and length, time from symptom onset and hospital referral, ongoing or recently COVID-19 infection. Outcomes measures were: death, amputation, and neurological deficit. Results: Between March 2021 and March 2022, 35 patients were surgically treated for PAA. Among them 15 referred to our hospital for symptomatic PAA and were urgently treated. Urgent treatments included both endovascular procedures and open surgeries. Nine out of 15 symptomatic patients had an ongoing or recently recovered COVID-19 infection. COVID-19 infection was strongly associated to symptoms development in patients affected by PAA and to surgical failure in those patients (OR 40, 95% CI 2.01–794.31, *p* = 0.005). Conclusion: In our series, presence of COVID-19 infection was strongly associated to ischemic symptoms onset and to complications after urgent treatment in symptomatic patients.

## 1. Introduction

The coronavirus SARS-CoV-2 (COVID-19) has spread widely since it was first identified and deeply transformed the vascular surgery clinical practice, causing delay in nonurgent and critical vascular issues during the outbreak period and decreased clinic, hospital, and emergency room consults.

In the same way, screening activities were, in most of the cases suspended or continued with reduced program [1].

In particular, the lockdown period had a significant impact on vulnerable patients resulting in a significant increase in vascular complication. The more significant influence has been recorded in peripheral arterial occlusive disease, resulting in an amputation rate increase in the first pandemic wave [2,3].

After this time, an increase in vascular procedures had been observed, without immediate return to pre-pandemic intervention rate, with total volume remaining 14% less when compared to the previous year [4]. In this context, vascular surgery, clinical, educational, and operative experiences were also limited [5].

After the widespread vaccine and the mild variants distributions, the vascular activities slightly came back to normal.

However, during this period every vascular surgeon had treated at least one patient affected by COVID-19 with thromboembolic events, most commonly acute limb ischemia or deep venous thrombosis [6,7]. The incidence of acute limb ischemia (ALI) increased, and the occurrences were mainly in patients who were COVID-19 positive [8]. In this patient’s cohort, surgical revascularization seems to be associated with a high technical failure rate, due to virus-related underlying hypercoagulability state. Some studies showed that abnormal coagulation results, characterized by elevated D-dimer level and fibrin degradation products, are common in patients affected by severe COVID-19 infection [9,10].

In fact, the hypercoagulability state in COVID-19 infection is often considered responsible for poor prognosis in affected patients and it can compromise results in urgent limb revascularizations. Furthermore, thrombotic effect of SARS-CoV-2 can linger long after recovery from infection, thus threatening surgical treatments. Despite the hypercoagulability state has been demonstrated to influence the incidence of acute limb ischemia, there are no data in the literature regarding the outcome of popliteal artery aneurysm (PAA) treatment in patients affected by COVID-19 infection.

The aim of this study is to report the incidence of symptoms in a series of patients affected by PAA treated in a tertiary hospital and to analyze how the ongoing or recent COVID-19 infection would affect the outcome in this cohort of patients.

## 2. Materials and Methods

We performed a single-center, observational cohort study. The data from all patients surgically treated for PAA, both in the elective and urgent setting, at the Vascular Surgery Unit of Policlinico Umberto I Hospital from the massive widespread of COVID-19 vaccine to present were collected in a prospectively maintained database. Data entry was care of the physicians involved directly with the patient.

For the present study, the data from all the patients who had presented with and been treated for PAA were identified and analyzed. The recorded variables included demographic data, comorbidities, medical and surgical history, operative details, and postoperative events during hospitalization and the immediate postoperative period. Factors considered for comparative analysis were: presence of PAA-related symptoms, aneurysm diameter and length, time from symptom onset and to hospital referral, ongoing or recently recovered COVID-19 infection (<2 months). Time from symptoms onset to hospital referral was evaluated for the present study and analyzed if arrived within 12 h or after 12 h from ischemic symptoms onset. Outcome measures considered for comparative analysis were: death, major or minor amputation, sensory or motor deficit in the affected limb. A subgroup analysis was performed in symptomatic patients according to the presence of ongoing or recently recovered COVID-19 infection. According to our standardized program, all patients with a clinical suspicion for PAA underwent lower limb DUS and, when the clinical suspicion was confirmed, a computed tomography angiography (CTA) to assess the extent of disease, the vessel size, and the run-off patency was performed.

Moreover, every patient underwent routine preoperative screening blood tests, including creatine phosphokinase and D-dimer, chest radiographs, and electrocardiography.

Procedural planning was carried out using the Osirix software ^TM^, with multiplanar and 3D reconstruction. Our routine approach to CT imaging is to perform multiplanar reconstruction as an essential point in the procedural design.

We defined symptomatic patients as those affected by PAA that arrived at our attention with complete aneurysmal thrombosis or with thromboembolic complication, while asymptomatic patients arrived at our attention for an incidental finding, without any ischemic symptoms.

Treatments performed in urgent settings were: fibrinolytic infusion, mechanical thrombectomy, aneurysm exclusion with endoprosthesis placement, and standard Fogarty embolectomy, while in the elective setting treatments were: aneurysm exclusion with endoprosthesis placement and/or femoro-popliteal bypass (Figure 1). In the urgent setting, the endovascular approach for thrombosed PAA was the first line of treatment in our center. All patients arrived for symptomatic PAA underwent mechanical thrombectomy using the Indigo Systems (Penumbra Inc., Alameda, CA, USA).

In presence of an adequate patent lumen, adequate run-off, and good landing zones, the placement of an endograft (Gore Viabahn ^TM^) was performed immediately after the mechanical thrombectomy. Nevertheless, in case of a suboptimal result after the mechanical thrombectomy, adjunctive fibrinolytic infusion was performed, and the stent placement was delayed to a good patent lumen achievement (Figure 2). The protocol for fibrinolytic infusion was intra-arterial locoregional low-dose Urokinase from 5000 to 7000 UI/hour, with associated unfractionated heparin infusion, for at least 48 h. Coagulation was tested every 6 h in patients under thrombolytic infusion. In case of a less than 80% basal level of antithrombin III, 500 U of antithrombin III was infused for 10 min before starting the continuous intravenous heparin to reach an activated partial thromboplastin time between 2 and 3 s. Whenever the endovascular approach was contraindicated or failed, a classic open surgical approach with a Fogarty distal embolectomy, (Le Maitre, Burlington, MA, USA) followed by a femoro-popliteal bypass were performed (Figure 3). Before arterial clamping, routinely 100 UI/kg of intravenous heparin and antibiotic prophylaxis were administered.

Fasciotomy was performed in the urgent setting whenever required because of compartment syndrome development risk. In the elective setting, the first choice in elder patients was the endovascular exclusion with endoprosthesis implantation, while the infragenicular femoro-popliteal bypass was the preferred route in younger patients.

Whenever an open surgical correction was deemed necessary, great saphenous vein was the conduit of choice. When no vein was available, heparin-bonded PTFE graft was implanted. Postoperative surveillance was performed with physical examinations and in symptomatic patients, full blood panel tests, every 6 h, for the first 48 h. Transfusions of packed red blood cells were given only if the hemoglobin level had decreased to less than 8 g/dL.

Technical success was defined both in the endovascular group and in the open group as the patency of the vascular endoprosthesis or prosthesis and distal run-off at least of one tibial vessel. Limb salvage was defined as the absence of major amputation in the treated leg.

At discharge, patients treated by endovascular surgery received dual antiplatelet therapy, while patients submitted to surgical bypass received single antiplatelet therapy.

All statistical analysis were conducted using the 27th version of SPSS (Statistical Package for the Social Sciences) (SPSS Inc., Chicago, IL, USA) and Excel (Microsoft Corporation, Redmond, WA, USA). Continuous variables were expressed as means ± standard deviation and categorial variables as proportions.

Kaplan Meier curves were used to calculate freedom from limb-related complications and major amputation-free survival. The Pearson chi-square test was used with nominal variables. *p*-value < 0.5 was considered statistically significant.

## 3. Results

Between March 2021 and March 2022, thirty-five patients were admitted and surgically treated for PAA at our Vascular Surgery Division. A total of 30 patients were male, 5 patients were female. The mean age was 74.86 years. Patient’s comorbidities are summarized in Table 1, with classical risk factors for atherosclerotic disease.

Among them, 15 referred to our hospital for symptomatic PAA, while the remaining 20 were asymptomatic aneurysms.

Among the symptomatic patients, one patient arrived at our center for symptomatic popliteal artery aneurysm and concomitant subarachnoid hemorrhage, with absolute neurosurgical contraindication to heparin administration so the symptomatic popliteal aneurysm was not treated. The patient died after 24 h for neurological complications. Twelve symptomatic patients received an endovascular treatment.

The mechanical thrombectomy was attempted in all of them. In ten patients, the thromboaspiration was successful, but in seven patients the result was suboptimal, so an infusion McNamara catheter was positioned, and the fibrinolytic infusion started. In seven patients, a stent implantation was required. In the remaining two patients, thromboaspiration was unsuccessful so those patients underwent a popliteal artery recanalization with subsequent endoprosthesis placement and associated fibrinolytic infusion.

Fogarty embolectomy was performed in two patients and after obtaining a good outflow vessel, the surgical femoro-popliteal bypass was performed, using an ePTFE prosthesis in one patient, and the reversed great saphenous vein in the other. Among elective treatment, covered-stent implantation was performed in 12 patients. Femoro-popliteal bypass was performed in eight patients, five with a prosthetic graft and three with reversed great saphenous vein.

Nine out of fifteen symptomatic patients had an ongoing or recently recovered COVID-19 infection (<2 months), while the remaining six were COVID-19 negative and had never been infected by COVID-19. Among the asymptomatic patients, 4 patients reported a recent (<2 months) COVID-19 infection, while the remaining 16 were COVID-19 negative and had never been infected by COVID-19.

In the whole cohort analyzed, the total number of patients infected by the virus was 13, 9 among symptomatic PAA and 4 among asymptomatic PAA; the remaining 22 patients were COVID-19 negative.

In symptomatic patients, time from symptoms onset to hospital referral was >12 h in 5, and <12 h in 10. Aneurysm diameter in the whole series was 35.6 ± 13.2 mm (mean ± SD), length was 175 ± 57.8 mm (mean ± SD). Post-operative complications were recorded in nine symptomatic urgent patients: major amputation in three cases, minor amputation in two cases, neurological motor deficit in two cases, neurological sensory deficit in one, and death in one case. In one patient, the motor deficit was associated with minor amputation. The total complication number were ten.

Complications developed in 9 patients out of 15 symptomatic (60%), while only 1 patient out of 20 recorded complications in the asymptomatic PAA (5%).

Among the complicated patients that were urgently treated, eight out of nine reported a recent COVID-19 infection, five referred to hospital >12 h from symptoms onset, six had a diameter ≥30 mm, and seven were more than 120 mm in length (Table 2). Among the patients treated electively, one blue toe syndrome due to distal embolism was developed after endovascular exclusion of PAA and the patient underwent a minor amputation. This patient has had a COVID-19 infection one month before.

Kaplan Meier analysis performed showed an overall freedom from limb-related complications at 7 days of 71.4%, of 95.5% in the COVID-19 negative population and of 30.8% in COVID-19 positive population. This difference was statistically significant (*p*-value < 0.001) (Figure 1).

In the asymptomatic and symptomatic PAA subgroup, the freedom from limb-related complications at 7 days was respectively of 90% and 46.7%, in the COVID-19 negative population respectively of 93.8% and 100% and in COVID-19 positive population respectively of 75.0% and 11.1% (*p*-value < 0.001) (Figure 2 and Figure 3).

In three patients, a major amputation was needed. In these cases, the revascularization of the thrombosed PAA was patent at the end of the procedure but the absence of run-off did not guarantee the patency of the graft and led to limb loss.

The Kaplan Meier analysis showed an overall major amputation-free survival at 7 days of 80%, of 95.5% in the COVID-19 negative population, and of 53.8% in COVID-19 positive population. This difference was statistically significant (*p*-value = 0.002) (Figure 4).

In the asymptomatic and symptomatic PAA subgroup, the major amputation-free survival at 7 days was respectively of 90% and 66.7%, in the COVID-19 negative population respectively of 93.8% and 100%, and in COVID-19 positive population of 75.0% and 44.4% (*p*-value < 0.011) (Figure 5 and Figure 6).

Overall technical success was 91.42%, technical success in the thrombosed PAA was 80%, while in the asymptomatic PAA was 100%. Limb salvage rate was 66.67% in the thrombosed PAA cohort; however, it is important to underline that the three patients who underwent a major amputation were COVID-19 positive, and so the limb salvage rate could be identified as rate in COVID-19 patient. In the whole cohort the limb salvage rate was 94.28%.

At statistical analysis, factors associated to post-operative complications were: presence of symptoms (OR 21.7, 95% CI 2.28–206.49, *p* = 0.001), ongoing or recently-recovered SARS-CoV-2 infection (OR 200, 95% CI 11.18–3576.89, *p* < 0.0001; Table 3 and Table 4). COVID-19 infection was strongly associated to symptoms development in patients affected by PAA (OR 28.5, 95% CI 2.97–273.31, *p* < 0.001), to complications in patients affected by PAA (OR 36, 95% CI 3.65–354.31, *p* = 0.00001), and to surgical failure in those patients (OR 40, 95% CI 2.01–794.31, *p* = 0.005).

Urgent treatment was strongly associated to complications in patients affected by PAA (OR 21, 95% CI 2.28–206.49, *p* < 0.001).

## 4. Discussion

The COVID-19 pandemic has generated deep consequences upon clinical care, workforce environments, and clinically generated revenue for nearly all vascular surgeons. Furthermore, the COVID-19 prevented most patients from hospital referral, for the infection fear, and as reported in the literature and by other colleagues this reality has been diffused across the world.

In the early pandemic period, our tertiary hospital was a dedicated COVID-19 center and our clinical activity was significatively diminished and only reduced to few emergent cases.

Due to this fact, we decided to analyze the period from the massive widespread of COVID-19 vaccine (i.e., March 2021 in Italy) to the first half of 2022.

As explained in the introduction section, the late pandemic period reported significantly less effects on vascular patients seen and treated, when compared to the first pandemic wave, with almost return to pre-COVID levels of cases, as reported by The VaSCular Activity Condition (VASCON) scale. This scale had been used as a way to describe the capability of hospitals/healthcare systems to provide surgical activity with limited resources, as happened during the pandemic era, especially during the first wave [2].

On the other hand, some epidemiological studies have encountered a significant increase in the incidence of ALI, when compared with the same period in pre COVID-19 era.

The viral infection has been associated with thromboembolic complications in both arterial and venous districts, due to hyperinflammation, platelet activation, endothelial dysfunction, and stasis advocated as predisposing factors for thrombotic complications [11]. The aggressiveness of COVID-19 pulmonary infection and the complexity of these patients have required minimally invasive approaches, in order to minimize the systemic burden of more aggressive vascular interventions and subsequent ICU need.

Despite the prothrombotic state of the viral infection has been associated with the augmented incidence of ALI, there are no data in the literature regarding the outcome of popliteal artery aneurysm (PAA) treatment in patients affected by COVID-19 infection.

The acute management of thrombosed PAA and subsequent ALI is a challenge even for an experienced vascular surgeon, due to the absence of a good tibial vessels run-off, that can lead to high peripheral resistance, early graft failure, and subsequent limb loss.

Thrombosed PAA with ALI can be treated both by open or endovascular intervention.

The current literature does not express a homogeneous consensus on the superiority of the endovascular strategy above the surgical route [12], despite some studies have underlined primary patency rate’s superiority in the endovascular procedures when compared to open surgery.

In fact, a metanalysis by Xiao et al. including 32 papers on thrombosed PAA highlighted an overall technical success rate of 90.73% without any significant difference between the open surgery and endovascular intervention groups, while the primary patency rate and the limb salvage rate at 1-year follow-up was significantly lower in the open group, when compared to the endovascular route [13].

On the other hand, a systematic review and meta-analysis showed that, compared with the endovascular approach, the open surgical approach was associated with higher primary patency at 1 year, lower embolic and thrombotic occlusions at 30 days, and lower reintervention rates [14].

In this contest, other studies are needed to understand the preferable route in thrombosed and non-thrombosed PAA; however, upon the range of procedural options, to address the critical issue of a disastrous run-off due to distal tibial and pedal arteries embolization, the strategy of initial thrombolytic treatment for ALI associated with PAA thrombosis has been proposed [15].

In this context, Dragas et al. have reported the use of intra-arterial thrombolysis has shown benefits in term of long term Major Adverse Limb Events (MALE) and overall survival, without significant risks of major bleeding complications.

The main focus in reducing the amputation rate is to reduce the time between ischemia onset and reperfusion, and in this context the mechanical thrombectomy devices reduce time to reperfusion significantly, i.e., within minutes and not hours.

In order to minimize the time of ischemia, the Penumbra/Indigo (Penumbra Inc.) aspiration thrombectomy system is a method of thrombus evacuation and has become our first line therapy in case of thrombosed PAA in order to achieve prompt reperfusion.

Furthermore, the presence of smaller aspiration catheters allows the vascular surgeon to reach the distal tibial vessels, often involved in the embolization of aneurysmal thrombus. In these cases, the endovascular route guarantees the possibility to adjunctive procedures such as the fibrinolytic infusion and covered stent implantation [16].

However, the use of the mechanical thrombectomy using the Penumbra thromboaspiration catheters in thrombosed PAA is reported only by few experiences in the literature.

In 2020, our group reported two cases of thrombosed popliteal artery aneurysm treated with immediate and direct revascularization using this device, that was proven to be safe and effective, allowing an immediate limb reperfusion, reducing at the same time the necessity for thrombolytic drug infusion [17].

On the other hand, De Donato et al. [18] have presented a standardized protocol of endovascular revascularization, based on the combination of vacuum-assisted thromboaspiration to improve tibio-pedal outflow and covered stent graft to exclude the PAA, reporting as expected preliminary encouraging 30-day results in terms of limb salvage.

In COVID-19 patients with acute arterial occlusion [19,20], there are a few cases reported in the literature that assess the catheter-directed thrombolysis to be a successful and safe option.

The necessity to understand which was the best management of ALI in the COVID-19 population emerged during this period, and this question was solved by a literature review in 2021, showing that the current ESVS recommendations remain valid or need adaptation COVID-19 population, due to the slightly changed epidemiology.

However, some minor modifications for organizational aspects were needed in order to minimize the need for ICU support, including the use of local or locoregional anesthesia, and endovascular techniques [21].

This review shows that the outcomes of patients with ALI seems to be worse in patients with simultaneous viral infection in term of mortality and amputation rates [22].

Another recent metanalysis by Galyfos et al. has enhanced the relationship between the viral infection and a higher risk for thrombotic complications, in fact patients with low incidence of comorbidities presented a virus-related limb ischemia [23].

The relationship between COVID-19 infection and hypercoagulability has been addressed in preliminary observations, as reported by Han et al. [5].

Upon the exact pathogenetic path of interaction, different proposed mechanisms have been hypothesized: on the one hand the virus affects the vascular endothelium through angiotensin-converting enzyme 2 (ACE-2) receptors, while on the other hand the prothrombotic state, leading to the thrombosis, is caused directly by the virus damaging the endothelial cells through alveolar ACE-2 receptors leading to endothelial cell activation and dysfunction [24]. A third hypothetic mechanism suggests that virus is the cause of an inflammatory cascade, that leads to a prothrombotic state and to microvascular and macrovascular endothelial damage [25,26].

The pathogenetic path and that the hypothetic mechanisms should be similar also in thrombosed popliteal artery aneurysm. However, the link between COVID-19 related arterial thrombosis and thrombosed popliteal artery aneurysms has not been described in the literature. On the contrary, more studies in this field are required to better understand whether upregulation of the angiotensin II/angiotensin II type 1 receptor proinflammatory axis by SARS-CoV-2 could influence aneurysm growth [27].

Despite the exact pathogenetic mechanism of interaction between the virus and the complication development is still unknown and should be clarified, the association between COVID-19 and thrombotic events seems to be stronger than the association existing between delayed hospital referral and complication development, as previously assessed in the pre-pandemic era.

However, despite reported series of acute limb ischemia after COVID-19, there are just a few case reports about popliteal artery aneurysm thrombosis in infected patients. To the best of our knowledge, this is the largest series of PAA treated during COVID-19 pandemic [28].

In fact, immediate diagnosis, accurate assessment, and urgent intervention are crucial to saving the limb and preventing a major amputation. Our experience underlines a more significant association between complications development and COVID-19 infection, than with time from symptoms onset [29].

In fact, comparing the results of our cohort of patients, it appeared clear that the limb salvage rate was diminished in comparison to the literature data on thrombosed PAA, with very poor outcome in term of major amputation-free survival and freedom from limb-related complications. As shown in the result section, and by the graphics, the overall rates in symptomatic patients were very poor, but they were poorer in the COVID-19 positive patients.

Moreover, the association between COVID-19 and the development of post-operative complications seems to be stronger than the association between ischemia-related-symptoms and complications and time from symptoms onset.

In fact, the statistical analysis assessed a strong association between the COVID-19 infection and the development of complication, both in the urgent and in the elective setting.

## 5. Conclusions

COVID-19 have wearied the entire medical community and in particular the vascular community has faced increased ischemic complications because of viral infection widespread.

In our series, the presence of ongoing or recently recovered infection was strongly associated with ischemic symptoms onset and with complications after urgent treatment in symptomatic patients and it has a predominant role in affecting surgical outcomes, despite the exact pathogenetic mechanism has not been established. Identification of ongoing or recent COVID-19 infections is of the outmost importance in surgical vascular patients.

Hopefully, general population should be fully aware of vascular COVID-19 complications and consider more timely vascular consults.

## Data Availability

Data is unavailable due to privacy and ethical restrictions.

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
