# Peer review of "Symptomatic Popliteal Artery Aneurysms in Recently SARS-CoV-2-Infected Patients: The Microangiopathic Thrombosis That Undermines Treatment"

_diagnostics, 2023, doi:10.3390/diagnostics13040647_

Round 1

Reviewer 1 Report

I would like to congratulate the authors on their work! This is potentially significant research regarding the outcome of PAA treated in a tertiary hospital in COVID-19 pandemic period. The authors conclusion underlying the negative impact of pandemic COVID-19 in the medical field, with a higher risk of ischemic events.

However, there are major aspects that must be addressed.

1.     It will be interesting if the authors presented the enrolled patients divided according to the present of COVID-19 infection at admission. Moreover, the Kaplan Meier plot based on the above-mentioned information are recommended.

2.     I suggest authors to improve the Introduction section with few references regarding the negative impact of pandemic COVID-19 on vascular surgery activity, as following:

- doi.org/10.3389/fsurg.2022.883935

- doi: 10.1016/j.hjc.2020.07.00

-https://doi.org/10.1016/j.jvs.2021.04.022

- https://doi.org/10.1016/j.jvs.2021.07.222

- https://doi.org/10.1177/1538574420985775

- https://doi.org/10.1016/j.avsg.2020.07.025

  3. It will be interesting if the authors compare their results and outcome of patients with PAA with the results of non-COVID-19 patients from the literature

Minor comments:

- please correct in all manuscript with “COVID-19”, there were several forms of writing “covid-19”, “covid19”

- please use in all the Table same font.

- The Discussion section must be rearranged, now it appears to be like some key idea presented in the different paragraph. The information’s is very pertinent, but the authors cand improved the aspect.

- English needs to be improved and typo errors corrected.

Author Response

Dear reviewer,

We would like to express our appreciation for your thoughtful response and the advice to our manuscript entitled “Symptomatic Popliteal Artery Aneurysms In Recently Sars-Cov2-Infected Patients: The Microangiopathic Thrombosis That Undermines Treatment”. We have carefully followed your suggestion and comments on the manuscript, so please find below our answers:

Point 1: It will be interesting if the authors presented the enrolled patients divided according to the present of COVID-19 infection at admission. Moreover, the Kaplan Meier plot based on the above-mentioned information are recommended.

Response 1: We divided patients as suggested by your comment and we mad K.M. plots based on that division. Please find that (in red) in the results section.

 Point 2: I suggest authors to improve the Introduction section with few references regarding the negative impact of pandemic COVID-19 on vascular surgery activity.

Response 2: many thanks for this helpfully suggestion, we tried to rearrange the introduction, including the papers indicated by you. Please find that in red in the introduction section.

Point 3:  3. It will be interesting if the authors compare their results and outcome of patients with PAA with the results of non-COVID-19 patients from the literature

Response 3: we report some literature results of PAA treatment in the preCOVID era (Line 428-436) and we tried to highlights the differences between the groups, and making in evidence the worse outcome of patients treated during COVID.

Point 4: minor comments,

Response 4: thanks again for giving su the opportunity to make our paper more fluent and acceptable, we made all changes suggested by you including the rearrangement of the discussion section.

Reviewer 2 Report

The authors report their experience on symptomatic PAAs in the Covid era. The aim of the paper is not clear. In addition, it is out of the context of the special issue (CLTI in the Covid era). And it is out of the aims of the journal (the paper reports surgical outcomes and nothing refers to diagnostics).

Maybe the authors should rewrite the paper adding some details about diagnostics (US findings in symptomatic PAA in the Covid era?)

Author Response

Dear Reviewer, I would like to thank you for your comments and advices regarding our manuscript entitled: Symptomatic Popliteal Artery Aneurysms In Recently Sars-Cov2-Infected Patients: The Microangiopathic Thrombosis That Undermines Treatment.

Response to the comments:

Regarding the aim of the paper, we thought to report our experience treating symptomatic PAA in the COVID era, and made in evidence the worse outcome of those patients compared to patients treated before COVID era reported in literature.

We add some details about the diagnostic process especially regarding the US which represent our first diagnostic approach in all patients (LINE 94-96).

Round 2

Reviewer 1 Report

no further comments

Reviewer 2 Report

Accept